# *Endarachne binghamiae* Ameliorates Hepatic Steatosis, Obesity, and Blood Glucose via Modulation of Metabolic Pathways and Oxidative Stress

**DOI:** 10.3390/ijms26115103

**Published:** 2025-05-26

**Authors:** Sang-Seop Lee, Sang-Hoon Lee, So-Yeon Kim, Ga-Young Lee, Seung-Yun Han, Bong-Ho Lee, Yung-Choon Yoo

**Affiliations:** 1Department of Microbiology, College of Medicine, Konyang University, Daejon 32992, Republic of Korea; wgd.aria@gmail.com (S.-S.L.); parkss2105@naver.com (S.-H.L.); sksthdus09@hanmail.net (S.-Y.K.); jc0012003@naver.com (G.-Y.L.); 2Department of Anatomy, College of Medicine, Konyang University, Daejon 32992, Republic of Korea; jjzzy@konyang.ac.kr; 3Department of Chemical and Biological Engineering, College of Engineering, Hanbat National University, Daejon 34158, Republic of Korea; lbh011@hanbat.ac.kr; 4CINNAM Ltd., Daejon 34158, Republic of Korea

**Keywords:** AMPK, *Endarachne binghamiae*, MASLD, obesity, steatosis

## Abstract

Obesity and metabolic dysfunction-associated steatotic liver disease (MASLD) are major contributors to the rise in metabolic disorders, particularly in developed countries. Despite the need for effective therapies, natural product-based interventions remain underexplored. This study investigated the therapeutic effects of *Endarachne binghamiae*, a type of brown algae, hot water extract (EB-WE) in ameliorating obesity and MASLD using high-fat diet (HFD)-induced ICR mice for an acute obesity model (4-week HFD feeding) and C57BL/6 mice for a long-term MASLD model (12-week HFD feeding). EB-WE administration significantly reduced body and organ weights and improved serum lipid markers, such as triglycerides (TG), total cholesterol (T-CHO), HDL (high-density lipoprotein), LDL (low-density lipoprotein), adiponectin, and apolipoprotein A1 (ApoA1). mRNA expression analysis of liver and skeletal muscle tissues revealed that EB-WE upregulated *Ampkα* and *Cpt1* while downregulating *Cebpα* and *Srebp1,* suppressing lipogenic signaling. Additionally, EB-WE activated brown adipose tissue through *Pgc1α* and *Ucp1,* contributing to fatty liver alleviation. Western blot analysis of liver tissues demonstrated that EB-WE enhanced AMPK phosphorylation and modulated lipid metabolism by upregulating PGC-1α and UCP-1 and downregulating PPAR-γ, C/EBP-α, and FABP4 proteins. It also reduced oxidation markers, such as OxLDL (oxidized low-density lipoprotein) and ApoB (apolipoprotein B), while increasing ApoA1 levels. EB-WE suppressed lipid peroxidation by modulating oxidative stress markers, such as SOD (superoxide dismutase), CAT (catalase), GSH (glutathione), and MDA (malondialdehyde), in liver tissues. Furthermore, EB-WE regulated the glucose regulatory pathway in the liver and muscle by inhibiting the expression of *Sirt1, Sirt4, Glut2*, and *Glut4* while increasing the expression of *Nrf2* and *Ho1*. Tentative liquid chromatography–tandem mass spectrometry (LC-MS/MS) analysis for EB-WE identified bioactive compounds, such as pyropheophorbide A and digiprolactone, which are known to have antioxidant or metabolic regulatory activities. These findings suggest that EB-WE improves obesity and MASLD through regulation of metabolic pathways, glucose homeostasis, and antioxidant activity, making it a promising candidate for natural product-based functional foods and pharmaceuticals targeting metabolic diseases.

## 1. Introduction

Metabolic dysfunction-associated steatotic liver disease (MASLD), formerly called non-alcoholic fatty liver disease (NAFLD), is a rapidly rising metabolic condition characterized by excessive hepatic fat accumulation, oxidative stress, and impaired lipid regulation [1,2]. MASLD is closely associated with obesity, dyslipidemia, cardiovascular disease, chronic kidney disease, and insulin resistance and has become a major global health burden without any FDA-approved therapeutic agents to date [3,4,5,6]. It encompasses a broad spectrum of liver diseases, including metabolic-associated steatosis (MASL) and metabolic-associated steatohepatitis (MASH) [5].

Signals driving MASLD progression are closely linked to oxidative mechanisms and disruptions in lipid and glucose metabolism [7]. At the molecular level, its pathogenesis involves dysregulation of metabolic genes, such as *Ampk*, *Pparγ*, *Srebp1c*, and *Fabp4* [8,9]. Among these, *Ampk* is regarded as a key metabolic regulator that regulates downstream targets, including the ACC–β-oxidation axis, FAS pathway, SIRT1–PGC-1α–UCP1 thermogenic pathway, and GLUTs. During MASLD, *Pparγ* expression is upregulated, synergizing with *Srebp1c* to promote FAS-mediated adipogenesis [10]. The ACC–CPT1 axis maintains the balance between fatty acid synthesis and oxidation, whereas the SIRT-1–PGC-1α–UCP1 axis regulates mitochondrial activity and energy expenditure. Excessive lipid accumulation chronically induces reactive oxygen species (ROS) generation through lipid peroxidation and FABP4 upregulation, which disrupts β-oxidation and redox equilibrium. On the defense side, NRF2 regulates redox homeostasis through antioxidant systems, including HO-1, G6PD, NADK, GSH, TRX, and PRDX. However, chronic metabolic dysregulation strengthens the link between metabolism, oxidative stress, and inflammation in MASLD–MASH progression, ultimately leading to the breakdown of these redox buffering systems [11].

In an era of increasing interest in preventive health care, seaweed has emerged as a promising natural resource with great potential for pharmaceutical development [12]. Among them, brown algae, along with red algae and green algae, have attracted increasing attention as functional candidates for improving metabolic diseases. Brown algae contain abundant bioactive polysaccharides—such as fucoidan, laminarin, and alginic acid—as well as diverse polyphenols [13]. These compounds have exhibited a variety of biological activities, including anticancer, anti-inflammatory, antioxidant, and anti-apoptotic effects. Recent studies have demonstrated that algae-derived compounds can attenuate hepatic steatosis by improving AMPK–SIRT-1 pathways and mitochondrial function and reducing lipid peroxidation. As an example, brown algae prevent MASLD by downregulating lipogenic genes and restoring metabolic signalings [12,13,14,15].

*Endarachne binghamiae* (EB) is a brown algae native to the Pacific coasts of Korea and Japan that has traditionally been used as a folk remedy for diabetes and inflammatory diseases [15]. EB is rich in many compounds with various physiological activities, such as phenols and diterpenoids, but it is still a little known brown algae. Recent studies have reported that EB has potential as a functional natural product to regulate adipogenesis in 3T3-L1 murine preadipocytes and protect against hydrogen peroxide (H_2_O_2_) induced oxidative stress in vitro [16,17]. These findings suggest that EB may contain functional compounds that can modulate lipid metabolism, redox signaling pathways, or both. However, there are no reports on the protective effect of EB on liver-related diseases caused by oxidative stress, such as MASLD. These facts led us to address whether the extract of EB (EB-WE) can ameliorate MASLD by restoring hepatic redox homeostasis and regulating lipid metabolism.

Since MASLD is a chronic inflammatory disease involving complex interactions among insulin resistance, lipid metabolism, and oxidative stress, comprehensive evaluation of interconnected factors in in vivo and in vitro systems is essential to verify the inhibitory activity of natural products for this disease [18]. Therefore, we validated the activity of EB-WE using a high-fat diet (HFD)-induced MASLD mouse model and evaluated hepatic steatosis, antioxidant capacity, lipid metabolism markers, and key signaling proteins. Here, in particular, we attempted to integrate in vivo physiological analyses with biochemical profiling of biomarkers, including redox-related molecules (GSH, GSSG, and CAT), adipogenesis regulators (p-ACC, FAS, SREBP-1c, and PPARγ), and redox metabolism proteins (SIRT and AMPK). In this study, we investigated the MASLD inhibitory effects of EB-WE in HFD-induced MASLD and obesity mouse models from the perspectives of anti-obesity, anti-diabetic, and antioxidant mechanisms.

## 2. Results

### 2.1. Tentative UHPLC-MS/MS Identification of Secondary Metabolites in EB-WE

Total ion chromatograms (TICs) of the EB-WE were obtained in both positive and negative electrospray ionization (ESI) modes, as shown in Figure 1. Peak deconvolution and compound annotation were carried out using high-resolution mass spectrometry (HRMS) data, MS/MS fragmentation, and in silico spectral matching.

A total of 14 secondary metabolites were tentatively identified at Level 2 according to the Metabolomics Standards Initiative (MSI) guidelines (Figure 1, Appendix A, Appendix A). These included two major candidates of biological relevance—pyropheophorbide A and digiprolactone—based on their well-matched MS/MS spectra, accurate mass, and marine occurrence. Other putative compounds identified included acanthoside K3, saikosaponin E, ephedradine C, kansuinin D, vachanic acid methyl ester, daturametelin H, 2-pentadecanone, trans-sobrerol, helveticoside, adenosine, allitol, and 6α-acetoxy-5-epilimonin. Their detailed identification parameters—such as monoisotopic mass, observed *m*/*z*, MS/MS spectra, mass accuracy, retention time, adducts, and in silico annotation results—are summarized in Figure 1C.

Notably, pyropheophorbide A exhibited four major diagnostic fragments (e.g., *m*/*z* 223.1, 355.1, 435.2, and 447.2) and matched the GNPS reference spectrum with a high in silico score (0.875). Digiprolactone (loliolide) was detected at two closely related retention times (5.96 and 6.43 min), both with consistent *m*/*z* and fragmentation patterns. Based on in silico data and previous literature, both peaks were considered meaningful features. A comprehensive summary of all identified compounds, including monoisotopic mass, MS/MS fragments, mass accuracy, retention time, and in silico annotations, is provided in Appendix A. Representative peak profiles obtained in positive and negative ion modes are shown in Appendix A, respectively. Furthermore, the detailed MetFrag scoring results and structural validation of the two key candidates—pyropheophorbide A and loliolide—using GNPS spectral matching and SIRIUS-based fragmentation trees are presented in Appendix A. The rationale for confidence level assignment of these two compounds, as well as the biological relevance of all 14 identified secondary metabolites, is discussed in detail below (Section 3).

### 2.2. Inhibition of Weight Gain by EB-WE in an Acute Obesity Model

To preliminarily evaluate the anti-obesity effects of EB-WE, we used a 4-week HFD-induced acute obesity model. Treatment with EB-WE significantly inhibited weight gain in a dose-dependent manner, showing higher efficacy than that of the positive control group (fenofibrate-treated) (Figure 2). The results of autopsy at 4 weeks revealed that abdominal and epididymal fat weights prominently increased in the HFD group but decreased in EB-WE-treated groups (Figure 2). Serological analyses showed that EB-WE (2 mg/mouse) significantly reduced TG, T-CHO, and LDL levels, approaching those of the ND group (Table 1).

### 2.3. Inhibition of Adipocyte Differentiation by EB-WE in 3T3-L1 Cells

The lipogenesis inhibitory activity of EB-WE was assessed using Oil Red O staining in 3T3-L1 cells treated with MDI (IBMX, dexamethasone, and insulin). EB-WE improved adipocyte differentiation by inducing AMPK phosphorylation (Figure 3).

### 2.4. Inhibitory Effect of EB-WE on Weight Gain in the MASLD Model

Based on the results in Figure 2, the dosage of EB-WE used in the MASLD model was determined to be 2 mg/mouse. Long-term oral administration of EB-WE for 12 weeks did not show any hepatotoxicity or nephrotoxicity (Appendix A), indicating that EB-WE had no toxic effect on the liver (GOT and GPT) and kidney (CRE and BUN). During 12 weeks of HFD feeding, body weight was measured weekly after fasting. The inhibitory effect of EB-WE on weight gain was apparent from week 4 and remained significant throughout the 12-week period (Figure 4A). Notably, the weight-reducing effect of EB-WE was higher than that of the positive control, sulforaphane.

To investigate the anti-obesity effect of EB-WE in more detail, the weights of abdominal and epididymal fat were measured at the end of the experiment. HFD feeding markedly increased the fat mass in both tissues, whereas, as expected, EB-WE significantly reduced their weights (Figure 4B). The fat-reducing efficacy of EB-WE was equal to or greater than that of the positive control, sulforaphane. In addition to fat mass reduction, histological analysis of abdominal adipose tissue by H&E staining revealed a significant reduction in adipocyte size in EB-WE-treated mice, suggesting that pathological white adipose tissue expansion was inhibited (Figure 4C). These results are consistent with the observed downregulation of adipogenic markers such as FAS and FABP4 at both protein and mRNA levels (Figure 5 and Figure 6), further confirming the inhibitory effect of EB-WE on white adipogenesis.

To explore whether enhanced thermogenic activity was associated with the reduction of fat mass by EB-WE, we assessed the expression of UCP-1, a key marker of brown adipose tissue (BAT) activation, in epididymal fat tissue using immunohistochemistry. As shown in Figure 4C, EB-WE prominently upregulated UCP-1 expression, indicating induction of browning programs. Similarly, histological examination of epididymal fat showed reduced adipocyte size, further supporting an increase in energy expenditure via thermogenesis. These findings collectively suggest that EB-WE exerts anti-obesity effects through dual mechanisms: inhibiting white fat hypertrophy and activating brown fat thermogenesis. As shown in the Western blot analysis presented in Section 2.6 (Figure 5), EB-WE also upregulated the expression of key thermogenic regulators (PGC-1α and UCP-1) in liver tissue. Furthermore, based on the liver tissue PCR results presented in Section 2.7 (Figure 6), it is possible that the whole-body energy consumption promotion effect by EB-WE was induced through the PGC-1α–UCP-1 axis. These findings collectively support that EB-WE exerts anti-obesity effects not only by reducing fat mass but also by enhancing fat browning and mitochondrial thermogenesis across multiple tissues.

### 2.5. Blood Profiles of Lipid-Related Indicators in the MASLD Model

Lipid-related serological parameters were analyzed to confirm the inhibitory effect of EB-WE on MASLD. The levels of T-CHO, TG, LDL, and ApoB were significantly decreased in the EB-WE-treated group compared to the HFD group, while adiponectin and ApoA1 were significantly increased (Table 2). In that ApoA1, the largest component of HDL, and ApoB, which reflects the amount of triglyceride-rich LDL, are meaningful factors in relation to metabolism, the levels of these molecules may explain the state of metabolic diseases like MASLD [19,20]. The ApoB–ApoA1 ratio, which represents the ratio of ApoA1 to ApoB, was observed as follows: ND, 1.97 ± 0.11; HFD, 0.766 ± 0.28; sulforaphane, 1.15 ± 0.29; and EB-WE, 1.23 ± 0.32. Considering that adiponectin, ApoA1, and ApoB are major predictors of MASLD [21], the results of serological analysis apparently indicated that EB-WE has an inhibitory effect on MASLD. Moreover, EB-WE was shown to have a positive effect on diabetes by controlling blood glucose levels.

### 2.6. Inhibition of Lipid Metabolism and Oxidative Stress by EB-WE in the Livers

Next, to address whether EB-WE can regulate the dysfunction of lipid metabolism in liver tissues, we investigated the effect of EB-WE on the expression of various molecules related to lipidogenesis and oxidative stress induced by HFD feeding in the livers. Treatment with EB-WE significantly reduced the weight of the livers as well as the levels of oxidation markers such as OxLDL and ApoB in liver tissues, but elevated the level of ApoA1, playing an important role in controlling lipid homeostasis (Figure 5A). Considering that OxLDL is an important indicator of oxidative stress generated during MASLD [22], EB-WE appears to suppress MASLD by regulating oxidation in liver tissue. Consistent with these results, EB-WE was found to upregulate the expression of molecules that inhibit oxidative stress superoxide dismutase (SOD), catalase (CAT), and glutathione (GSH)) but downregulate the concentration of malondialdehyde (MDA), a marker of lipid peroxidation produced by oxidative stress in the liver (Figure 5B). These findings suggest that EB-WE can alleviate lipid peroxidation by exerting a metabolic regulatory mechanism based on antioxidants such as GSH. This suggestion was visually supported by frozen sections of liver tissue stained with Oil Red O (Figure 5C). Western blot analysis using liver tissues also showed that EB-WE enhanced the phosphorylation of AMPK, which protects against diet-induced obesity and NAFLD [7] (Figure 5D). The enhancement of AMPK phosphorylation by EB-WE in liver tissue was the same as that observed in 3T3-L1 cells in Figure 3C. In addition, treatment of EB-WE increased the expression of PGC-1α and UCP-1 regulatory proteins for lipid metabolic homeostasis and suppressed the expression of PPAR-γ, C-EBP-α, and FABP4 proteins that promote lipidogenesis [8] (Figure 5E). Meanwhile, from a redox regulation perspective, EB-WE was shown to ameliorate oxidative stress in liver tissues by upregulating antioxidant proteins, NRF-2 and HO-1 (Figure 5E).

**Figure 5 ijms-26-05103-f005:**
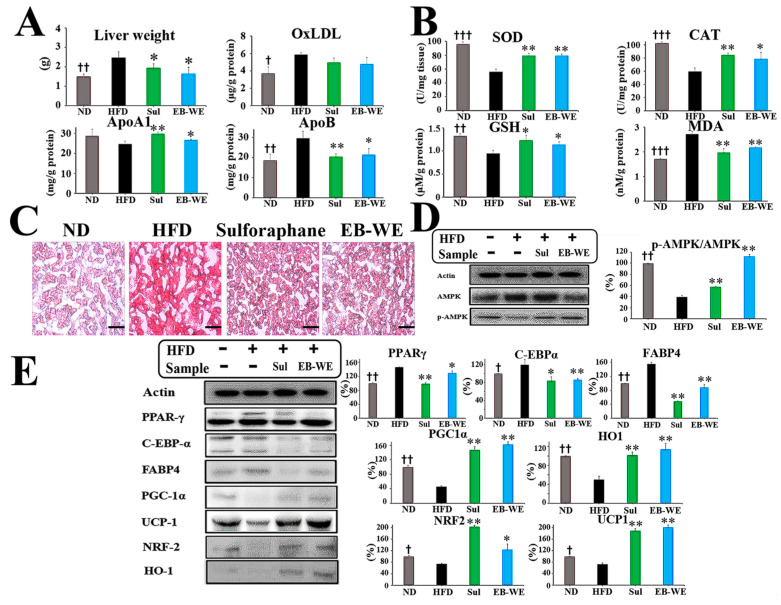
Inhibitory effect of EB-WE on lipid metabolism and oxidative stress in liver tissues. (**A**) Liver tissue weight, ApoA1, ApoB, and OxLDL levels. (**B**) SOD, catalase, GSH, and MDA levels. (**C**) Liver tissue images using Oil Red O staining, scale bar, 50 µm. (**D**,**E**) Western blot analysis using liver tissues. Values are expressed as mean ± SD. * *p* < 0.05; ** *p* < 0.01, compared with the HFD group. ^†^ *p* < 0.05; ^††^ *p* < 0.01; ^†††^ *p* < 0.001, (ND vs. HFD) compared with the HFD group. Abbreviations: OxLDL, oxidized low-density lipoprotein; SOD, superoxide dismutase; GSH, reduced glutathione; MDA, malondialdehyde; PPARγ, peroxisome proliferator-activated receptor gamma; C-EBPα, CCAAT/enhancer-binding protein alpha; FABP4, fatty acid-binding protein 4; PGC-1α, peroxisome proliferator-activated receptor gamma coactivator-1 alpha; UCP-1, uncoupling protein 1; NRF2, nuclear factor erythroid 2–related factor 2; HO-1, heme oxygenase-1.

### 2.7. Regulation of mRNA Expression in Relation to Lipid Metabolism, Glucose Metabolism, and Oxidation in Liver Tissues

The results in Table 2 and Figure 5 suggested the potential of EB-WE to regulate oxidative stress, glucose metabolism systems, and lipid-related metabolism. Based on these results, we carried out further investigation to define the effect of EB-WE on the expression of mRNAs associated with lipid metabolism, antioxidation, and glucose metabolism in liver tissues using real-time PCR. EB-WE was shown to efficiently regulate the expression of many types of mRNAs associated with metabolic processes to improve MASLD (Figure 6). That is, for genes related to lipid metabolism, EB-WE significantly downregulated *Pparγ*, *Srebp1*, *Fas*, *Cebpα*, and *Fabp4* gene expression but upregulated *Pgc-1α* gene expression. The expression of AMPK mRNA (*Ampkα1* and *Ampkα2*) was increased by EB-WE. Since AMPK is an upstream regulator of these lipid metabolism–related genes, the inhibitory effect of EB-WE on lipid metabolism–related gene expression may be due to upregulation of AMPK genes. Regarding glucose metabolism, *Sirt1* and *Glut2* gene expression were upregulated by EB-WE. Furthermore, EB-WE has been observed to upregulate the expression of *Nrf2*, *Ho1*, and *Nqo1*, consistent with the protein-level results presented in Figure 5. This upregulation suggests that EB-WE may activate the Nrf2 signaling pathway, which plays a crucial role in cellular defense mechanisms against oxidative stress. These findings suggest that the anti-obesity and MASLD improvement effects of EB-WE appear to be achieved through regulation of upstream signaling pathways that control these genes in the liver.

**Figure 6 ijms-26-05103-f006:**
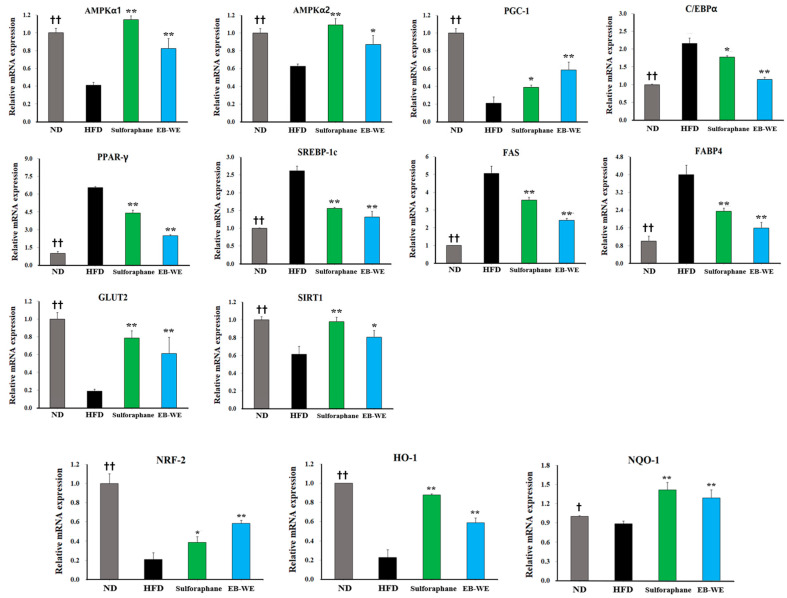
Regulation of metabolic gene expression by EB-WE in liver tissues. The expression of mRNAs related to lipid metabolism (*Pparγ*, *Srebp1c*, *Fas*, *Cebpα*, *Fabp4*), glucose metabolism (*Glut2*, *Sirt1*), and antioxidation regulation (*Nqo1*, *Ho-1*, *Nrf2*) was analyzed by real-time PCR. Values are expressed as mean ± SD. * *p* < 0.05; ** *p* < 0.01, compared with the HFD group. ^†^ *p* < 0.05; ^††^ *p* < 0.01 (ND vs. HFD), compared with the HFD group. Abbreviations: *Ampkα1*, AMP-activated protein kinase alpha 1 subunit; *Ampkα2*, AMP-activated protein kinase alpha 2 subunit; *Glut2*, glucose transporter type 2; *Nqo1*, NAD(P)H:quinone oxidoreductase 1; *Fas*, fatty acid synthase; *Srebp1c*, sterol regulatory element-binding protein 1c; *Sirt1*, sirtuin 1.

### 2.8. Regulation of mRNA Expression in Skeletal Muscle Tissues

In the same manner as the experiment of Figure 6, the expression of major metabolic mRNAs was examined in skeletal muscle tissues obtained 12 weeks after HFD feeding (Figure 7). Expectably, EB-WE significantly upregulated the expression of *Ampk*, *Cpt1a*, *Glut4*, and *Sirt4* mRNAs. These results confirm that EB-WE’s effects on obesity and metabolic disease arise from its comprehensive regulation of gene expression related to metabolic pathways not only in the liver but also in skeletal muscle.

### 2.9. Improvement of Circulating Glucose Regulation by EB-WE in MASLD Model

MASLD often accompanies impaired fasting blood sugar or type 2 diabetes, and dysregulated glucose homeostasis may be a major factor in its progression [23]. As shown in Table 2, EB-WE treatment significantly lowered blood glucose concentrations at week 12 of HFD feeding in the MASLD model. To further evaluate whether EB-WE improves glucose tolerance under metabolic challenge, we conducted an oral glucose tolerance test (OGTT) at week 10, as described previously [24]. EB-WE treatment significantly suppressed the rapid rise in blood glucose levels observed at 15 and 30 min after glucose intake and caused glucose levels to return to near-normal range more quickly compared to untreated mice (Figure 8). These results suggest that EB-WE may help control glycemia and maintain systemic glucose homeostasis in the MASLD model.

### 2.10. Inhibitory Effect of EB-WE on Lipid Accumulation in an In Vitro MASLD Model

The inhibitory effect of EB-WE on fat accumulation in liver tissues was examined in an in vitro MASLD model in which HepG2 hepatocytes were treated with free fatty acids (FFAs, oleic acid, and palmitic acid) and stained using Nile red staining method. FFAs induced the increase of intracellular lipid vacuoles in HepG2 hepatocytes, but treatment of EB-WE apparently reduced the formation of lipid vacuoles in a dose-dependent manner (Figure 9A). The inhibitory effect of EB-WE on lipid accumulation in HepG2 cells was confirmed by the decreased concentration of intracellular triglyceride (TG) in the cells (Figure 9B). This fact strongly proposed that EB-WE can suppress the lipid accumulation induced by FFAs even in nonadipose cells such as hepatocytes.

## 3. Discussion

### 3.1. Activation of AMPK and Suppression of Lipogenesis

Dysregulation of lipid metabolism, redox balance, and glucose homeostasis is closely associated with MASLD pathogenesis [25].

AMPK, a heterotrimeric complex consisting of a catalytic α-subunit and regulatory β- and γ-subunits, restores energy balance by increasing ATP production through decreased anabolic processes and enhanced catabolism. AMPK acts as a central regulator of metabolic homeostasis and can be activated by various factors such as mitochondrial inhibition, nutrient deprivation, and exercise [26,27]. Ca^2+^/calmodulin-dependent protein kinase II (CAMKII) and liver kinase B1 (LKB1) are upstream regulators involved in AMPK activation via the phosphorylation of Thr172. LKB1 mediates AMPK activation primarily under mitochondrial dysfunction and energy stress, while CAMKII can directly activate AMPK at Thr172 in response to calcium flux [28,29]. AMPK participates in several biosynthetic pathways, including regulating lipid and sterol metabolism by inhibiting acetyl-CoA carboxylase 1 and 2 (ACC1 and ACC2) and 3-hydroxy-3-methylglutaryl-coenzyme A reductase (HMGCR), which is involved in cholesterol synthesis [30]. Since AMPK maintains metabolic homeostasis through the regulation of various downstream pathways [30], activation of this molecule is considered a promising therapeutic target for many diseases associated with excessive lipid synthesis, such as obesity, hyperlipidemia, and fatty liver disease [31]. In this study, EB-WE was shown to enhance AMPK activity in the liver and muscle tissues in the MASLD mouse model, suggesting that AMPK activation is a key factor in its anti-MASLD effect (Figure 5, Figure 6 and Figure 7).

Based on this finding, we further demonstrated how EB-WE ameliorates MASLD by regulating AMPK activity and its downstream pathways, especially from the perspective of lipid metabolism.

Peroxisome proliferator-activated receptors (PPARs) consist of three subtypes (α, β, and γ) and serve as key regulators of energy metabolism in various tissues, including the liver, skeletal muscle, adipose tissue, and intestine [32]. PPAR-γ promotes lipogenesis through interactions with CCAAT/enhancer-binding protein (C/EBP) family members and sterol regulatory element-binding protein (SREBP) family members [33,34,35].

As shown in Figure 5 and Figure 6, treatment of EB-WE dramatically suppressed lipogenesis in liver tissues by downregulating PPAR-γ, C/EBP-α, and SREBP-1c, thereby reducing fatty acid synthase (FAS) and fatty acid binding protein 4 (FABP4) activity [36]. Given that these transcriptional regulators are modulated downstream of AMPK activation, the observed lipogenesis-inhibiting effects are likely mediated, at least in part, through AMPK signaling.

### 3.2. Induction of Fat Browning and CPT-1-Mediated Fatty Acid Oxidation

Brown adipose tissue plays a critical role in obesity suppression by increasing energy expenditure through thermogenesis [37,38,39]. In the process of lipid metabolism, peroxisome proliferator-activated receptor gamma coactivator-1 alpha (PGC-1α) mediates brown adipogenesis by upregulating uncoupling protein 1 (UCP-1). As shown in Figure 4, Figure 5 and Figure 6, administration of EB-WE induced fat browning by upregulating the PGC-1α–UCP-1 pathway, and this lipogenesis-inhibiting and browning-inducing effect of EB-WE was supported by the downregulation of fatty acid binding protein 4 (FABP4), a transporter of fatty acids [40,41,42]. Moreover, its effect on brown adipogenesis was confirmed through histological results from liver, abdominal fat, and epididymal fat staining (Figure 4 and Figure 5). The expression of CPT-1 is negatively correlated with hepatic triglyceride accumulation and blood lipid levels [43] while positively correlated with fatty acid oxidation and improved glucose homeostasis in skeletal muscle [44]. In the skeletal muscle of high-fat diet-fed mice, EB-WE significantly increased CPT-1 expression (Figure 7). This suggests that upregulation of CPT-1 is a mechanism related to the enhancement of energy metabolism by EB-WE.

### 3.3. Regulation of Glucose Metabolism via SIRT–GLUT Axis

The sirtuin family, consisting of seven members (SIRT1–7), plays a crucial role in inhibiting MASLD and other liver diseases [45]. Among them, SIRT-1 is a major molecule regulating lipid and glucose metabolism through deacetylation of NF-κB and P53 [45]. Additionally, SIRT-1 has been reported to regulate brown adipogenesis and mitochondrial function via PGC-1α, influencing the development of MASLD or MASH [46]. The glucose transporter (GLUT) family, comprising 14 members (GLUT1–14), is responsible for transporting glucose from the extracellular space into the cytoplasm, playing a pivotal role in glucose homeostasis [47]. EB-WE upregulated the expression of *Sirt1* and *Glut2* in the liver (Figure 6) and *Sirt4* and *Glut4* in skeletal muscle (Figure 7). Considering the well-established functional link between mitochondrial regulation and glucose homeostasis across tissues [48], the upregulation of the *Pgc-1α*–UCP-1 pathway by EB-WE, as discussed in Section 3.2, may be mediated, at least in part, through *Sirt1* and *Sirt4*.

### 3.4. Antioxidant Response via NRF2 and Redox Enzymes

In the progression of MASLD, oxidative stress induced by lipid peroxidation, mitochondrial dysfunction, and endoplasmic reticulum stress are key factors [49,50]. In this context, redox regulation plays an important role in the onset and progression of MASLD and other metabolic diseases [51]. Clinical and statistical studies have reported that antioxidant activity may contribute to the prevention of MASLD and metabolic diseases [52,53]. Here, we found that EB-WE increased the levels of antioxidant enzymes such as SOD, CAT, and GSH but reduced the level of MDA, a compound derived from oxidized unsaturated fatty acids, in liver tissue (Figure 5). Regarding the oxidation reaction, the NRF-2–HO-1 pathway is a critical antioxidant mechanism and has been reported to contribute to MASLD improvement [54]. EB-WE also upregulated the expression of both NRF-2 and HO-1 antioxidant proteins in liver tissue (Figure 5 and Figure 6).

### 3.5. Tentative Identification of Key Bioactive Compounds

As described in Section 2.1 and Section 4.3, based on an integrated evaluation—including marine occurrence, mechanistic bioactivity evidence, and high-confidence structural matching via MS/MS—we selected pyropheophorbide A and digiprolactone (loliolide) as the most plausible bioactive candidates in EB-WE.

Pyropheophorbide A, an intermediate metabolite of chlorophyll, is likely to be isolated from marine algae, including brown algae [55,56]. This is because pyropheophorbide A can be converted from pheophytin a during the degradation process of chlorophyll a, which is abundant in brown algae along with chlorophyll c. In fact, a study reported the antioxidant activity of the green algae *Enteromorpha prolifera* after ethanol extraction and partial fractionation, with pyropheophorbide A suggested as the main component mediating this activity [57]. Another study reported antioxidant activity in the brown algae *Cladostephus spongiosus*, where pyropheophorbide A was identified as having high antioxidant potential along with fucoxanthin [58]. Furthermore, a study on the activity of pyropheophorbide A detected in the green algae *Capsosiphon fulvescens* reported that this compound inhibits endothelial dysfunction mediated by advanced glycation end products (AGEs) [59]. Indeed, key intermediate metabolites involved in AGE formation, such as MGO and GO, tend to increase in fasting glucose disorders and type 2 diabetes associated with abnormalities in AMPK-based metabolic systems in the liver and are implicated as causes of chronic oxidative damage related to lipid peroxidation and ferroptosis pathways [60]. This compound has been reported with the same monoisotopic mass in multiple studies [61,62], with some also showing nearly identical retention times. Taken together with the high similarity in MS/MS fragmentation patterns, its frequent occurrence in marine algae, and numerous reports of its bioactivity related to metabolic improvement, this compound was therefore determined to correspond to a high-confidence Level 2 metabolite identification.

Loliolide, a monoterpene lactone, has been identified in various seaweeds, particularly within the Sargassum genus. For instance, it has been isolated from *Sargassum ringgoldianum* subsp. coreanum and *Sargassum horneri* [63,64]. Additionally, loliolide has been detected in other marine algae, including *Undaria pinnatifida* and *Prasiola japonica*, indicating its widespread distribution across diverse seaweed species [65,66]. Beyond its presence in seaweeds, loliolide exhibits notable bioactivities pertinent to metabolic health. Studies have demonstrated its antioxidant properties, effectively scavenging reactive oxygen species (ROS) and enhancing cell viability under oxidative stress conditions [67]. Furthermore, loliolide has shown anti-inflammatory effects by reducing pro-inflammatory cytokine production and modulating key signaling pathways, such as MAPK and NF-κB [68]. In the context of adipogenesis, loliolide suppresses lipid accumulation and downregulates adipogenic transcription factors, including PPARγ, C/EBPα, and SREBP1c, while activating AMPK [69].

These combined activities suggest that loliolide may contribute to metabolic improvement by mitigating oxidative stress, reducing inflammation, and inhibiting adipocyte differentiation [70,71]. Furthermore, loliolide has been shown to upregulate NAD(P)H quinone oxidoreductase 1 (NQO1) expression, aligning with the findings from our qRT-PCR analysis. Specifically, studies have demonstrated that loliolide activates the Nrf2/HO-1 signaling pathway, leading to increased expression of antioxidant enzymes, including NQO1 [69]. Given that the compound analysis results were in excellent agreement with previously reported data—including MS/MS fragmentation patterns and monoisotopic mass as documented in databases such as HMDB (HMDB0302428)—and showed retention times within 1 min under similar chromatographic conditions, these compounds are considered to be high-confidence Level 2 identifications under the MSI framework. Although more reliable analyses are needed for definitive confirmation, it is likely that the properties of the EB-WE components described above contribute to its biological activity. Therefore, further studies involving definitive compound isolation and structural elucidation are planned to confirm these tentative identifications and clarify their contribution to the observed bioactivities.

### 3.6. Functional Relevance of Additional Tentative Compounds

In addition to the primary candidates, we also investigated other putatively identified compounds through literature-based evaluation of their biological relevance. Azedarachtin has been reported to exhibit anti-inflammatory effects, contribute to mitochondrial function recovery, and regulate glutathione homeostasis [72]. Ephedradine, a spermine-derived alkaloid, has demonstrated antihypertensive properties and may contribute to metabolic improvement [73]. Vachanic acid, also known as ilicic acid, has been proposed to inhibit ectopic lipid accumulation [74] and has also been reported to exhibit potential anti-inflammatory activity [75]. Kansuinin D, a coumarinolignan, has limited evidence of occurrence in seaweeds but has been reported to exert anti-apoptotic and anti-atherosclerotic effects through inhibition of ROS generation and downregulation of the IKKβ–IκBα–NF-κB pathway [76]. Daturametelin H, a withanolide-type steroidal lactone, has been suggested to possess antioxidant and anti-inflammatory activities [77]. 2-Pentadecanone, a C15 aliphatic ketone, has also been reported to exhibit antioxidant properties [78]. Helveticoside, a steroid-based cardiac glycoside, has been suggested as a potential therapeutic agent for MASLD (metabolic dysfunction-associated steatotic liver disease) [79], and Allitol, a hexitol, has shown anti-obesity activity in previous studies [80,81]. These mechanistic similarities provide additional support for the plausibility of the tentative identifications.

Notably, a large number of the tentatively identified compounds in EB-WE appeared to be either terpenoid derivatives or structurally related to terpenoid precursors. Saikosaponin E, a triterpene saponin, was found to improve sphingolipid metabolism via apolipoprotein E regulation [82]. 6α-Acetoxy-5-epilimonin, a limonoid triterpenoid, has limited reports of marine origin but has been suggested to possess antioxidant potential and 5α-reductase inhibitory activity [83]. Likewise, Acanthoside K3, a triterpenoid glycoside, has demonstrated a wide range of bioactivities, including suppression of tAGE–RAGE signaling, inhibition of NLRP3-mediated pyroptosis in ulcerative colitis, anti-inflammatory effects in LPS-stimulated RAW 264.7 macrophages, and neuroprotective effects in scopolamine-induced amnesia models via activation of the TrkB–CREB–BDNF pathway [84,85,86]. Despite these mechanistic similarities, most triterpenoid-related compounds have been rarely reported in seaweed species. However, some studies have isolated such compounds from microalgae, including Mychonastes sp. 247 [87]. Additionally, in our previous study conducted within our laboratory [88], EB-WE significantly alleviated acute lung injury in vivo and suppressed TLR4–MyD88 signaling in LPS-stimulated RAW 264.7 macrophages in vitro. Based on these findings, we consider the possibility that triterpenoid-related compounds may either be synthesized by the seaweed itself or by endogenous microbial symbionts present in the seaweed matrix.

## 4. Materials and Methods

### 4.1. Reagents

The PCR primers used in the study were purchased from Cosmo Gentech (Seoul, Republic of Korea). Fetal bovine serum (FBS), RPMI1640, and penicillin/streptomycin were purchased from WENGENE (Gyeongsan, Republic of Korea), and DNA polymerase was purchased from Bio-Rad (Hercules, CA, USA). Reagents other than those mentioned above were purchased from Sigma-Aldrich (St. Louis, MO, USA).

### 4.2. Preparation of EB-WE

EB-WE used in the study was extracted from Hanbat University (Daejeon, Republic of Korea), as described previously [88]. Briefly, finely ground EB (50 g) was mixed with 10 times the volume of distilled water (500 mL) and extracted at 80 °C for 6 h. The supernatants obtained through filtration and centrifugation were then lyophilized using a freeze dryer (VaCo2, Niedersachsen, Germany) at −83 °C. The hot water extract of EB (EB-WE) was stored at −30 °C until use.

### 4.3. UHPLC-MS/MS

To tentatively identify the major constituents of EB-WE, we performed UHPLC analysis (UHPLC/1290 Infinity; Agilent, Santa Clara, CA, USA) coupled with Q-TOF mass spectrometry (G6550A; Agilent). Several compounds were annotated with MSI Level 2 confidence based on MS/MS fragmentation and database matching, as presented in Figure 1. Chromatographic separation was performed using an ACQUITY UPLC^®^ HSS T3 column (1.8 µm, 2.1 × 100 mm; Waters, Milford, MA, USA) maintained at 35 °C. The mobile phase consisted of water (solvent A) and acetonitrile (solvent B), both containing 0.1% formic acid. The gradient elution was as follows: 3% B at 0–1 min, linear ramp to 100% B at 25 min, held until 27 min, and returned to 3% B by 30 min, with a total run time of 35 min. The flow rate was 0.5 mL/min, and the injection volume was 5 µL. Mass spectrometric detection was carried out on a Waters SYNAPT G2-Si Q-TOF mass spectrometer equipped with an electrospray ionization (ESI) source, operated in both positive and negative ion modes. The capillary voltage was set at 3.0 kV (ES+) or 2.5 kV (ES−), with a source temperature of 100 °C and desolvation temperature of 250 °C. The desolvation gas flow was 500 L/h (ES+) or 600 L/h (ES−), and cone gas flow was 50 L/h. MS data were acquired over an *m*/*z* range of 100–1200 at a resolution of 20,000. MS/MS data were collected using a data-independent acquisition (DIA) mode with ramped collision energy (20–45 eV for ES+, 25–50 eV for ES−).

To enhance annotation confidence, we employed a multi-platform strategy integrating spectral library matching, in silico fragmentation, and fragment tree analysis. MS/MS spectra were first compared against the GNPS spectral library [89] to identify known compounds based on experimental fragmentation patterns. Additionally, MetFrag-based in silico fragmentation was performed using five curated databases (PubChem, KEGG, HMDB, ChEBI, and MetaCyc) to evaluate structural plausibility and match quality [90]. Fragment tree prediction and structural consistency were further assessed using SIRIUS [91]. Chromatogram deconvolution and peak selection were performed based on *m*/*z* error (<5 ppm), fragment coverage, and match scoring. Final candidate compounds were selected for further analysis, as presented in Section 2.1.

### 4.4. Obesity Model in Mice

In the high-fat diet (HFD)-induced acute obesity model, ICR mice (4-week-old males) purchased from Raon Bio (Yongin, Republic of Korea) were randomly divided into 4 groups of 10 each: Normal diet (ND), HFD, Fenofibrate, HFD+EB-WE 1 mg/mouse, and HFD+EB-WE 2 mg/mouse. EB-WE and fenofibrate were orally administered once daily via gavage at a volume of 200 μL/mouse for 4 weeks. ICR mice exhibit rapid weight gain (20–25% increase within 4 weeks on HFD) due to their hyperphagic nature and accelerated lipid absorption [92,93]. This short-term model was used to assess early adipogenesis and serum lipid responses to EB-WE, in contrast to long-term focus such as the MASLD model [94]. HFD used in this study was based on the D12492 formulation consisting of 60% kcal of fat, 20% kcal of carbohydrate, and 20% kcal of protein (Research Diets, Inc., New Brunswick, NJ, USA). Normal diet (ND) group received low-fat control diet (D12450K, 10% kcal from fat, no sucrose; Research Diets). Body weight was measured once a week at the same time of day after an 8-h fast. All animals had free access to food and water throughout the experimental period. The detailed macronutrient composition and ingredient list are shown in Appendix A. Obesity animal experiments using ICR mice was carried out in accordance with the guidelines of the Institutional Animal Care and Use Committee (IACUC) of Konyang University (Approval No. P-22-12-E-01).

### 4.5. HFD-Induced MASLD Model and Oral Glucose Tolerance Test (OGTT)

MASLD model was performed using C57BL/6J mice (Raon Bio). C57BL/6J mice require a long-term period of >12 weeks to develop characteristic symptoms of MASLD induced by high-fat diet feeding, including hepatic steatosis, elevated fasting blood glucose levels (>140 mg/dL by week 10), and low-grade inflammation [92,93]. This strain has a genetic defect in leptin signaling that leads to progressive lipid accumulation in both adipose tissue and liver, reflecting a multifactorial pathogenesis similar to human MASLD [94,95]. Groups of 10 C57BL/6J mice (8-week-old males) were randomly divided into 4 groups: ND, HFD, HFD + sulforaphane (1.25 mg/mouse), and HFD + EB-WE (2 mg/mouse). Sulforaphane (Santa Cruz, CA, USA), a potent antioxidant with demonstrated anti-MASLD activity, was used as the positive control [96]. The HFD used in this model was the same formulation as in the ICR experiment (D12492, Research Diets) and was administered continuously for a total of 12 weeks to induce MASLD pathology. The Normal diet (ND) group received a matched low-fat control diet (D12450K, Research Diets). All animals had free access to food and water throughout the experimental period.

EB-WE and sulforaphane were dissolved in sterile phosphate-buffered saline (PBS) and administered once daily via oral gavage at a volume of 200 μL/mouse. The same vehicle (PBS, 200 μL) was administered to the ND and HFD control groups to ensure consistency across all experimental groups. A fixed-dose approach was used (not weight-adjusted), as all mice were of similar age and baseline weight at the initiation of treatment. In the case of the MASLD model using C57BL/6J mice, this design was considered appropriate given that metabolic changes in this strain typically occur gradually over time, as noted in the first paragraph. Additionally, each mouse was housed individually in a single cage, minimizing environmental variability and further justifying the use of a standardized dose. Body weight and food intake were measured weekly after an 8-h fasting period. OGTT was performed at week 10 of HFD feeding using the D12492 diet formulation. Blood glucose was measured at 15, 30, 60, 120, and 180 min after glucose administration. Mice were sacrificed at week 12, and the weights of epididymal fat, abdominal fat, liver, kidney, spleen, and pancreas were recorded.

All animal procedures were approved by the Institutional Animal Care and Use Committee of Konyang University (Approval No. P-24-13-A-01) and conducted in accordance with institutional and national guidelines for the care and use of laboratory animals.

### 4.6. Serological Analysis

In obesity and MASLD models, blood specimens obtained from HFD-fed mice at the indicated weeks were analyzed for various serological parameters. Blood lipid profiles—including total cholesterol (T-CHO), triglycerides (TG), HDL cholesterol (HDL), and LDL cholesterol (LDL)—as well as glucose levels were measured using a standard biochemical analyzer (Dri-Chem 3500s, FUJIFILM, Tokyo, Japan). Serum concentrations of adiponectin (µg/mL), apolipoprotein A1 (ApoA1, ng/mL), and apolipoprotein B (ApoB, ng/mL) were determined using ELISA kits from Promega (Madison, WI, USA). Serum levels of glutamate oxalate transaminase (GOT/AST), glutamate pyruvate transaminase (GPT/ALT), creatinine (CRE), and blood urea nitrogen (BUN) were also measured for preliminary testing for toxicity. The toxicity-related results are summarized separately in Appendix A.

Hepatic analyses included the measurement of antioxidant enzymes, including superoxide dismutase (SOD), catalase (CAT), and glutathione (GSH), which were quantified using DoGenBio colorimetric assay kits. Additionally, liver samples were analyzed for ApoA1, ApoB, and oxidized LDL (OxLDL) using Promega kits, while malondialdehyde (MDA) levels were also measured using DoGenBio assay kits.

### 4.7. Histological and Immunohistochemical Analysis

Frozen liver, epididymal fat, and abdominal fat tissues obtained from mice in the HFD-induced MASLD model were collected at the end of the experiment. For liver tissue, the largest hepatic lobe (typically the left lobe) was consistently used for histological analysis to minimize interindividual variability. Liver samples were fixed with 4% paraformaldehyde, dehydrated in 30% sucrose, embedded in OCT compound, and cryosectioned at a thickness of 7 μm using a CM1520 cryostat (Leica Biosystems). Sections were stained with Oil Red O to evaluate hepatic lipid accumulation.

Epididymal fat (representing visceral adipose tissue) and abdominal fat (primarily perirenal and peritoneal regions) were weighed separately, washed with PBS, and fixed in 4% paraformaldehyde for 2 h. After dehydration in 30% sucrose and embedding in OCT compound, 10 μm frozen sections were prepared. Abdominal fat was stained with hematoxylin and eosin (H&E) to assess white adipocyte hypertrophy. Epididymal fat sections were used for immunohistochemistry: after blocking with 3% BSA, sections were incubated overnight at 4 °C with anti-UCP-1 primary antibody (Abcam; 1:500). After washing, HRP-conjugated anti-rabbit IgG secondary antibody (Abcam; 1:1000) was applied for 30 min at room temperature, followed by DAB substrate (Abcam, Cambridge, UK) and hematoxylin counterstaining to visualize UCP-1 expression. The results of histological analyses and associated tissue weights were shown in Figure 4B (fat weight), Figure 4C (fat histology), and Figure 5A (liver histology and weight).

### 4.8. Western Blot

Liver tissues obtained at 12 weeks in the MASLD model and MDI-treated 3T3-L1 cells were lysed with RIPA buffer, and proteins (20 µg) were subjected to SDS-PAGE. Following transfer to PVDF membranes, the blots were incubated with specific primary and HRP-conjugated secondary antibodies. Protein bands were visualized using a chemiluminescence imaging system (Azure Biosystems, Dublin, CA, USA). The list of primary antibodies used for Western blot and immunohistochemistry, including clone information, catalog numbers, and the biological significance of each target protein, is summarized in Appendix A. The selected proteins represent key regulators of metabolic homeostasis (e.g., AMPKα and PPARγ), mitochondrial function and lipid oxidation (e.g., PGC-1α and UCP1), adipogenesis (e.g., C/EBPα and FABP4), and oxidative stress response (e.g., Nrf2 and HO-1).

### 4.9. Real-Time PCR

The expression of genes related to lipid and sugar metabolism and antioxidant activity was analyzed by real-time PCR (CFX96; Bio-Rad, Hercules, CA, USA) using liver tissues and skeletal muscles obtained from mice 12 weeks after HFD feeding in the MASLD model. The sequences of all primers used are listed in Appendix A.

### 4.10. Adipogenic Differentiation

Adipogenesis and lipid accumulation were assessed using 3T3-L1 preadipocytes (ATCC). 3T3-L1 preadipocytes were seeded into 6-well plates at a density of 2 × 10^5^ cells per well and cultured for 48 h. Once the cells were grown confluently, the culture medium was replaced with differentiation medium consisting of DMEM supplemented with 10% FBS, 1% P/S, and MDI (0.5 mM 3-isobutyl-1-methylxanthine, 1 μM dexame-thasone, and 10 μg/mL insulin) to induce adipogenic differentiation for 2 d. After the initial differentiation phase, the medium was replaced with DMEM supplemented with insulin or the indicated concentrations of EB-WE, and the cells were further incubated for an additional 3 d. At the end of the incubation period, Oil Red O staining was performed to assess lipid accumulation. Additionally, triglyceride (TG) level in the cells was measured using a TG assay kit (Asan Pharmaceutical, Seoul, Republic of Korea) at the indicated time points.

### 4.11. In Vitro MASLD Assay

HepG2 hepatocarcinoma cells were seeded into a 6-well plate at a density of 1 × 10^6^ cells/mL and cultured in DMEM supplemented with 10% FBS and 1% P/S for 24 h. After incubation, the culture medium was replaced with fresh DMEM containing 0.2 mg/mL of the test sample, ascorbic acid, and 1 mM free fatty acids (FFA; 0.5 mM oleic acid + 0.5 mM palmitic acid), followed by an additional 24-h incubation. At the end of the treatment period, the medium was removed, and the differentiated HepG2 cells were washed twice with PBS to remove any residual substances. Lipid accumulation was then assessed by staining the cells with Nile red (AdipoRed; Ecocell, Hanam-si, Gyeonggi-do, Republic of Korea), and the condition of lipid accumulation was observed using a fluorescence microscope.

### 4.12. Statistical Analysis

All statistical analyses were performed using IBM SPSS Statistics 25 and SAS 9.4 software, employing Student’s *t*-test or ANOVA methods.

## 5. Conclusions

In this study, we comprehensively evaluated the therapeutic potential of *Endarachne binghamiae* hot water extract (EB-WE) in the context of obesity and metabolic-associated steatotic liver disease (MASLD). Using a high-fat diet (HFD)-induced MASLD mouse model, EB-WE administration significantly reduced body weight, hepatic steatosis, and circulating glucose levels. These physiological improvements were accompanied by modulation of key metabolic pathways, including suppression of lipogenesis and enhancement of antioxidant defenses.

EB-WE activated AMPK, a central regulator of energy metabolism, and downregulated lipogenic transcription factors such as PPARγ, C/EBPα, and SREBP-1c. It also upregulated antioxidant response elements, including NRF2, HO-1, and NQO1, contributing to the reduction of oxidative stress. Furthermore, EB-WE improved serum lipid profiles (triglycerides, total cholesterol, and LDL cholesterol) and reduced fat accumulation in liver and adipose tissues while enhancing glucose tolerance.

The simultaneous improvements observed across diverse metabolic and redox pathways may, at least in part, be attributed to the upstream regulatory role of AMPK, supporting its function as a master integrator of EB-WE’s multifaceted effects. Collectively, these findings suggest that EB-WE exerts its beneficial effects through coordinated regulation of AMPK-centered metabolic pathways, involving lipid metabolism, glucose homeostasis, and antioxidant activity, as illustrated in Figure 10. Our results highlight EB-WE as a promising marine-derived therapeutic candidate for managing metabolic liver diseases and support its further investigation in clinical and mechanistic studies focused on its bioactive components.

## Figures and Tables

**Figure 1 ijms-26-05103-f001:**
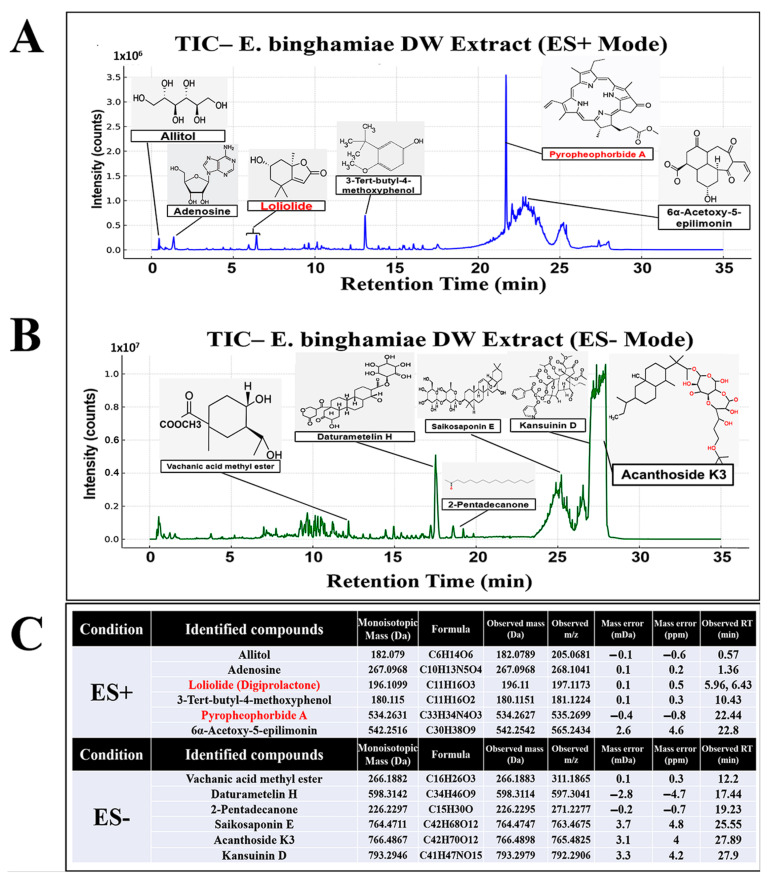
LC-MS/MS-based identification and in silico fragmentation analysis of major compounds from EB-WE (**A**,**B**) total ion chromatograms (TICs) of EB-WE obtained by UHPLC-QTOF-MS/MS in positive (ES+) and negative (ES−) ionization modes. The profiles include 14 representative compounds, with their peak positions and corresponding chemical structures shown in each mode. (**C**) Summary table of identified or putatively annotated compounds in both ionization modes. Two main compounds—pyropheophorbide A and loliolide (digiprolactone)—are highlighted in red due to their strong MS/MS match and literature validation in marine algae.

**Figure 2 ijms-26-05103-f002:**
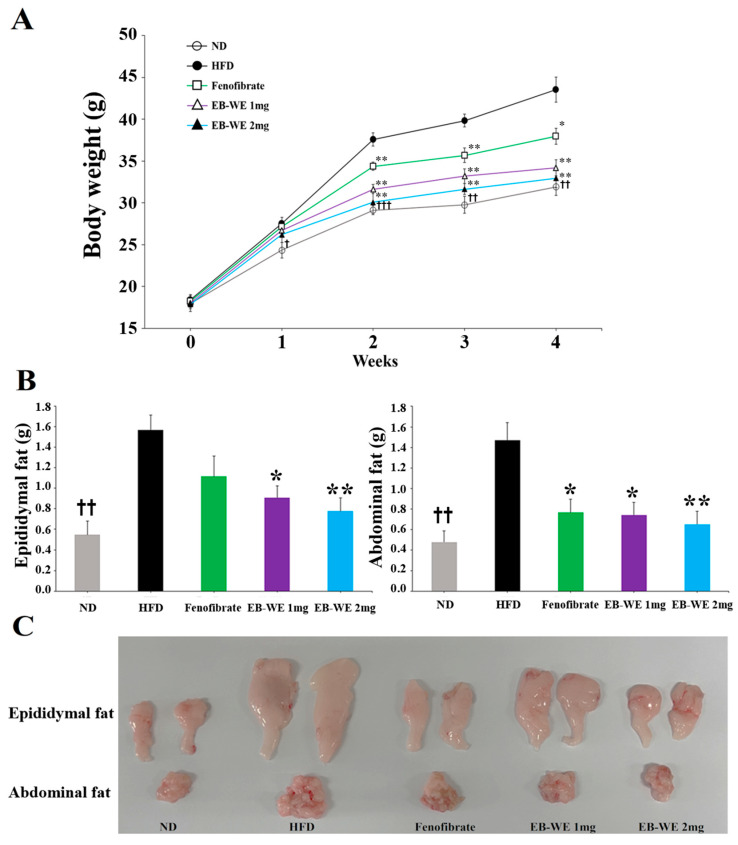
Anti-obesity effect of EB-WE in an HFD-induced acute obesity model. (**A**) Weight gain suppression. (**B**) Abdominal and epididymal fat weight. (**C**) Photographs of fat tissues. Values are expressed as mean ± SD. * *p* < 0.05; ** *p* < 0.01, compared with the HFD group. ^†^ *p* < 0.05; ^††^ *p* < 0.01; ^†††^ *p* < 0.001, (ND vs. HFD) compared with the HFD group. ND, Normal diet group; HFD, High-fat diet group; Fenofibrate, HFD + Fenofibrate (2 mg/mouse) group; EB-WE 1 mg, HFD + EB-WE (1 mg/mouse) group; EB-WE 2 mg, HFD + EB-WE (2 mg/mouse) group. Values are expressed as mean ± SD. * *p* < 0.05, compared with the HFD group.

**Figure 3 ijms-26-05103-f003:**
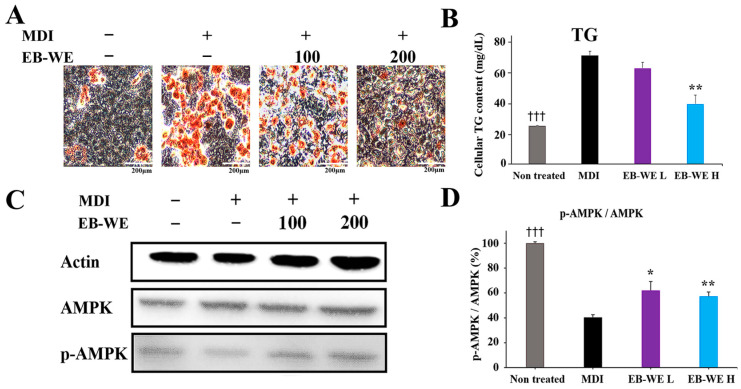
Inhibition of lipid differentiation by EB-WE in 3T3-L1 cells. (**A**) Oil Red O staining after lipid differentiation. (**B**) Quantification of TG. (**C**) Western blot analysis of AMPK phosphorylation. (**D**) Quantification of the p-AMPK/AMPK ratio. Values are expressed as mean ± SD. * *p* < 0.05, compared with the MDI group; ** *p* < 0.01, compared with the MDI group. ^†††^ *p* < 0.001, (ND vs. MDI) compared with the MDI group. Abbreviations: MDI, IBMX + dexamethasone + insulin (differentiation induction mix); AMPK, AMP-activated protein kinase; p-AMPK, phosphorylated AMPK.

**Figure 4 ijms-26-05103-f004:**
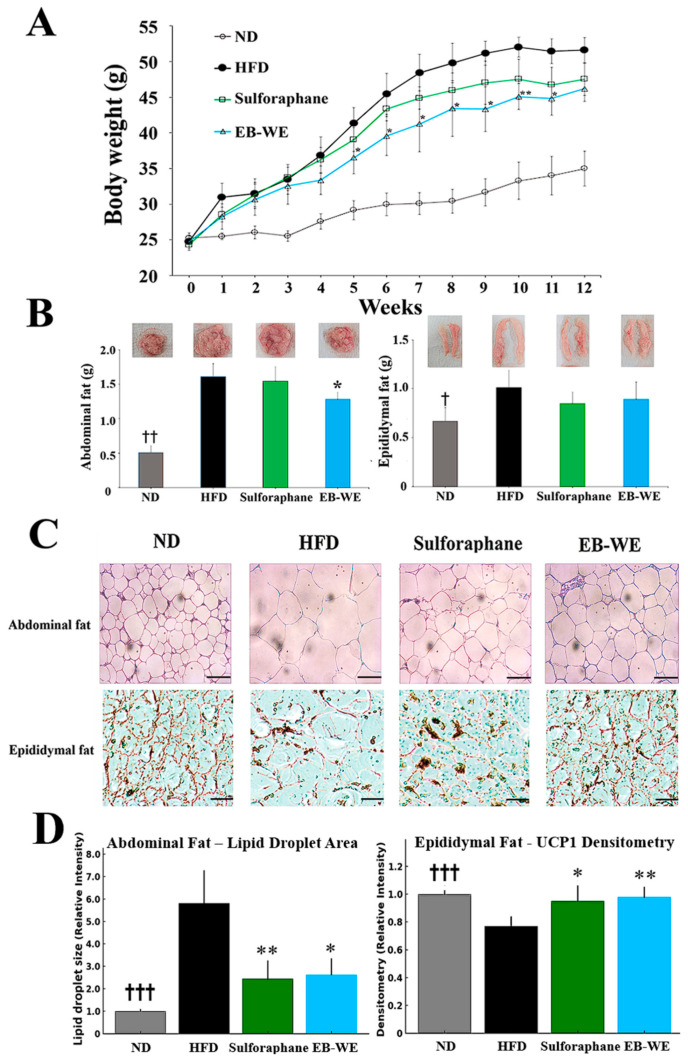
Inhibitory effect of EB-WE on weight gain and fat accumulation in the MASLD model. (**A**) Weekly changes in body weight. (**B**) Weights of abdominal and epididymal fat at week 12. (**C**) Representative histological images of adipose tissues, scale bar, 50 µm. (**D**) Quantification of lipid droplet area in abdominal fat (left) and densitometric analysis of UCP-1 staining in epididymal fat (right) based on the images shown in (C). Bar graphs adjacent to the images indicate quantification of adipocyte area (upper) and UCP-1–positive area (lower), respectively. Values are expressed as mean ± SD. * *p* < 0.05; ** *p* < 0.01, compared with the HFD group. ^†^ *p* < 0.05; ^††^ *p* < 0.01; ^†††^ *p* < 0.001, (ND vs. HFD) compared with the HFD group. Abbreviations: UCP-1, uncoupling protein 1.

**Figure 7 ijms-26-05103-f007:**
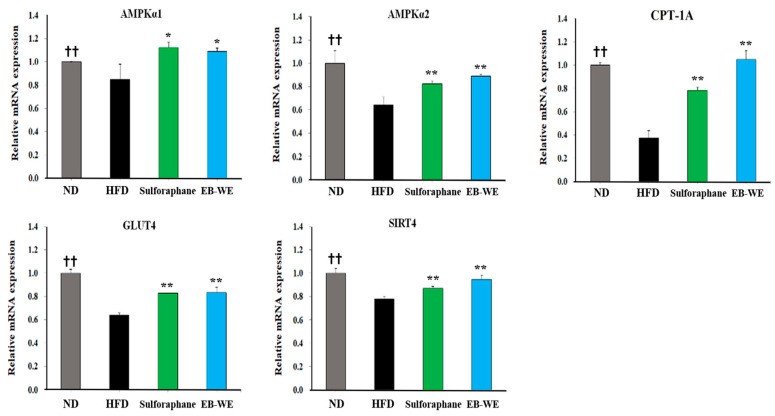
Regulation of metabolic gene expression by EB-WE in muscle tissues. The expression of mRNAs related to metabolic pathways (*Ampk*, *Cpt1a*, *Glut4*, *Sirt4*) was analyzed by real-time PCR. Values are expressed as mean ± SD. * *p* < 0.05; ** *p* < 0.01, compared with the HFD group. ^††^ *p* < 0.01 (ND vs. HFD), compared with the HFD group. Abbreviations: *Ampk*, AMP-activated protein kinase; *Cpt1a*, carnitine palmitoyltransferase 1A; *Glut4*, glucose transporter type 4; *Sirt4*, sirtuin 4.

**Figure 8 ijms-26-05103-f008:**
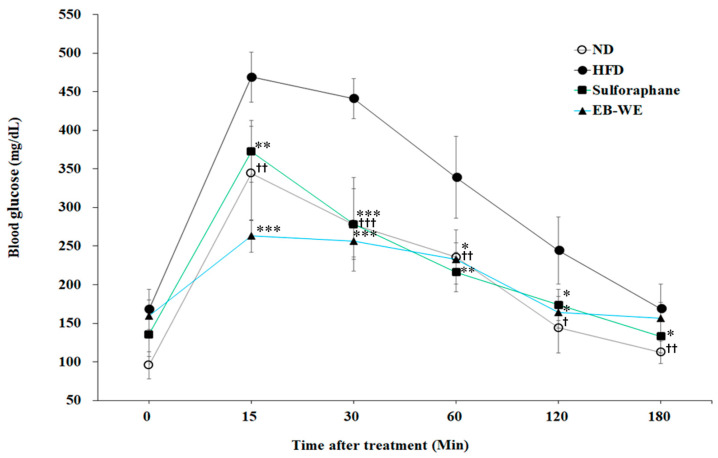
Effect of EB-WE on circulating glucose response in the OGTT. OGTT (oral glucose tolerance test) was carried out 10 weeks after HFD feeding. Values are expressed as mean ± SD. * *p* < 0.05; ** *p* < 0.01; *** *p* < 0.001, compared with the HFD group. ^†^ *p* < 0.05; ^††^ *p* < 0.01; ^†††^ *p* < 0.001, (ND vs. HFD) compared with the HFD group.

**Figure 9 ijms-26-05103-f009:**
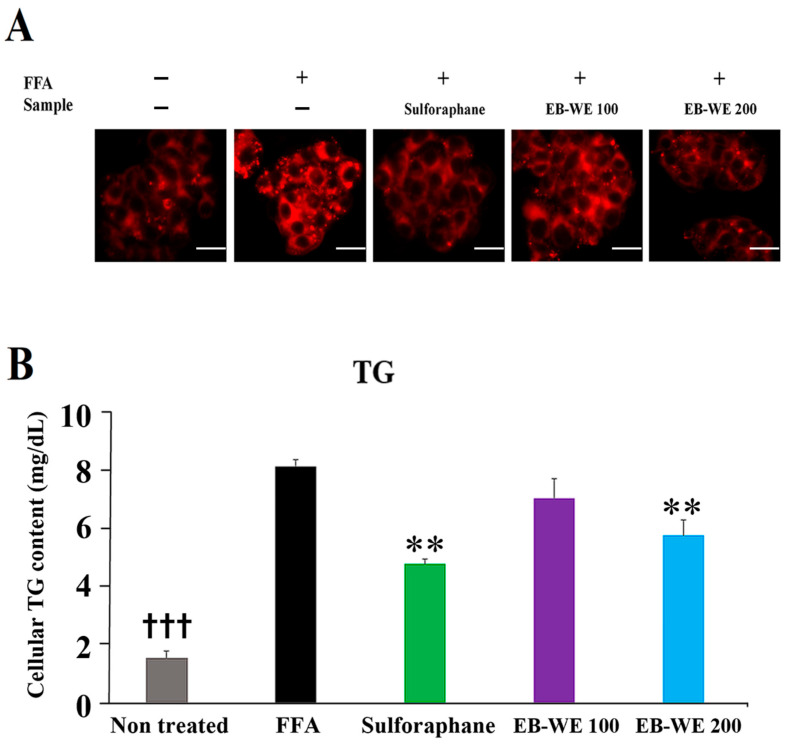
Inhibition of lipid accumulation by EB-WE in FFAs-treated HepG2 hepatocytes. (**A**) Nile red staining of HepG2 cells. scale bar, 25 µm (**B**) Intracellular level of TG. Values are expressed as mean ± SD. ** *p* < 0.01 compared with the FFAs-treated group. ^†††^ *p* < 0.001, (ND vs. HFD) compared with the HFD group. Abbreviations (new): FFAs, free fatty acids.

**Figure 10 ijms-26-05103-f010:**
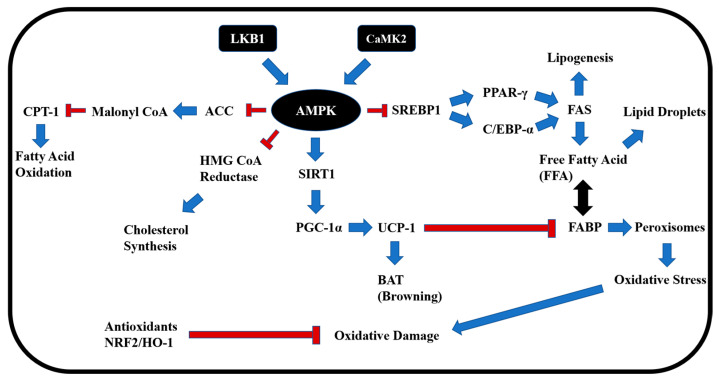
A schematic diagram illustrating AMPK-centered metabolic and redox regulation interactions. EB-WE is thought to improve MASLD through AMPK upregulation, which leads to the inhibition of lipogenesis, promotion of browning, antioxidant activity, and enhancement of basal metabolic rate. Abbreviations (new): ACC, acetyl-CoA carboxylase; LKB1, liver kinase B1; CaMK2, calcium/calmodulin-dependent protein kinase II; BAT, brown adipose tissue; HMG CoA reductase, 3-hydroxy-3-methylglutaryl-CoA reductase.

**Table 1 ijms-26-05103-t001:** Lipid metabolism-related serological parameters measured 4 weeks after HFD feeding.

Profiles	ND	HFD	Fenofibrate	EB-WE 1 mg	EB-WE 2 mg
TG (mg/dL)	107 ± 5 ^†^	139 ± 22	123 ± 5 *	127 ± 15	105 ± 6 *
T-CHO (mg/dL)	129 ± 15 ^††^	275 ± 33	222 ± 23	201 ± 32 *	185 ± 13 *
HDL (mg/dL)	82 ± 6 ^††^	110 ± 0	107 ± 4	110 ± 0	101 ± 6 *
LDL (mg/dL)	21 ± 19 ^†††^	137 ± 35	91 ± 26	73 ± 33 *	71 ± 19 *

EB-WE administration resulted in significant improvements in lipid metabolism-related serological parameters compared to the HFD group. Values are expressed as mean ± SD. * *p* < 0.05, compared with the HFD group; ^†^ *p* < 0.05; ^††^ *p* < 0.01; ^†††^ *p* < 0.001, (ND vs. HFD) compared with the HFD group. Abbreviations: TG, triglyceride; T-CHO, total cholesterol; HDL, high-density lipoprotein cholesterol; LDL, low-density lipoprotein cholesterol.

**Table 2 ijms-26-05103-t002:** Serological profiles measured 12 weeks after HFD feeding.

Profiles	ND	HFD	Sulforaphane	EB-WE
TG (mg/dL)	103.4 ± 11.4 ^††^	177.6 ± 16.8	136.3 ± 13.8 *	117.6 ± 12.8 **
T-CHO (mg/dL)	117 ± 13.0 ^†††^	212.4 ± 10.7	190.2 ± 9.7 *	176.3 ± 28.7 *
HDL (mg/dL)	100.9 ± 8.0 ^†^	110 ± 0	107.5 ± 3.9	109 ± 2.2
LDL (mg/dL)	10.5 ± 6.6 ^††^	66.6 ± 10.5	55.1 ± 9.5	51.7 ± 8.4
Adiponectin (ug/mL)	31.3 ± 1.0 ^††^	12 ± 0.2	21.8 ± 1.6 **	21.4 ± 0.2 **
ApoA1 (ng/mL)	47.5 ± 2.6 ^†††^	29.9 ± 1.3	36.6 ± 3.3 *	38.3 ± 4.0 **
ApoB (ng/mL)	24.1 ± 0.5 ^†††^	39 ± 0.4	31.7 ± 0.9 **	31.1 ± 1.3 **
Glucose (mg/dL)	95.7 ± 18.0 ^†††^	205 ± 15.0	135 ± 21.5 **	142 ± 17.0 **

Treatment of EB-WE significantly improved the levels of serological parameters related to lipid metabolism. Values are expressed as mean ± SD. ND, Normal diet group; HFD, HFD group; Sulforaphane, HDF + sulforaphane (1.5 mg/mouse) group; EB-WE, HFD + EB-WE (2 mg/mouse) group. * *p* < 0.05; ** *p* < 0.01, compared with HFD group. ^†^ *p* < 0.05; ^††^ *p* < 0.01; ^†††^ *p* < 0.001, (ND vs. HFD) compared with the HFD group. Abbreviations: HDL, high-density lipoprotein cholesterol; LDL, low-density lipoprotein cholesterol; ApoA1, apolipoprotein A1; ApoB, apolipoprotein B.

## Data Availability

Data are contained within the article and the Appendix A.

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
