# Peer review of "Endarachne binghamiae Ameliorates Hepatic Steatosis, Obesity, and Blood Glucose via Modulation of Metabolic Pathways and Oxidative Stress"

_ijms, 2025, doi:10.3390/ijms26115103_

Round 1

Reviewer 1 Report

Comments and Suggestions for Authors
  1. Introduction: Please further enrich the introduction, such as the hypothesis proposed against the research background, the proposed research method and the research significance. The references are not novel enough, and the discussion and summary of cutting-edge reports should be increased.
  2. The abbreviations used in the tables and figures should be explained in the footnote of tables and/or figure captions to make them standalone.
  3. Figure1, 3 and 5 should be given as clear picture. The font size of the data in some figures is too small for easy reading.
  4. The ND group should also be analyzed for significance against the HFD group.
  5. Keep the number of decimal places in the table uniform.
  6. Section 4.4 and 4.5: The components of a high-fat meal should also be provided.
  7. Animal experiments: Please provide more information related to animal experiments, including the feeding conditions of experimental animals, and ethical approval number or proof of ethical approval documents, etc.
  8. Different disease models selected mice of different ages and intervened for different periods of time. What is the basis?
  9. Review and revise reference styles according to journal requirements. 

Author Response

Thank you very much for taking the time to carefully review our manuscript. We have addressed all comments from Reviewer 1, 2, and 3 in a consolidated point-by-point response, organized by reviewer. Please refer to the attached file for detailed explanations and corresponding revisions.

Reviewer 2 Report

Comments and Suggestions for Authors

The present study investigated the potential of an aqueous extract of Endarachne binghamiae to improve parameters related to obesity, circulating glucose and oxidative stress. In addition to the impressive results in all these parameters, the authors evaluated the compounds present in the extract by LC-MS/MS and verified metabolic pathways using techniques such as Western blotting and real-time PCR. The results are highly impressive; however, the methodology requires refinement to enable other researchers to effectively replicate and build upon the findings.

Title: Replace 'insulin resistance' with 'circulating glucose'.

Abstract: cite the main compounds identified through the LC-MS/MS analysis

Reorganize the results section by presenting the epididymal and abdominal fat weights, followed by the photograph and analysis of brown adipose tissue

It would be interesting to include results such as animal growth, diet intake, total body weight, and overall body weight gain in animal models.

2.10 Replace all instances of 'insulin resistance' with 'circulating glucose'. Was this analysis performed after 10 or 12 weeks of HFD feeding?

4.2 Why was hot water used for the extraction? Wouldn’t the high temperature risk degrading important compounds? Were most of the important compounds associated with fibers?

4.4 and 4.5 Was EB-WE administered by gavage or incorporated into the diet? How were the intakes of water, diet, EB-WE, fenofibrate, and sulforaphane controlled during the study? The analysis should include a detailed examination of lipid profiles, weight gain, and the quantification of epididymal and abdominal fat. 

line 551 - "10 weeks"?

4.6 - where are GOT, GPT, CRE and BUN in results?

4.7 and 4.8 - Which specific part of the tissue was analyzed, and what was its corresponding weight?

4.8 - what about abdominal fat?

4.9 - Could you please specify the antibodies used and the proteins analyzed in the study?

4.11 - Was HepG2 used for this analysis?

Include in methodology: liver weight, TG, CHO, adiponectin, Apos, glucose, LDLox, SOD, CAT

A conclusion section should be included to summarize the main findings and their significance  

Author Response

(The authors gave the same response as above.)

Reviewer 3 Report

Comments and Suggestions for Authors

Abstract

  • Add a brief introduction related to the state of art
  • Please specify the model used

Introduction

  • Line 54: Please rephrase this sentence “most advanced seaweed system” the complexity of brown algae is not at all the more advanced also other algae species belonging from different categories are as well “advanced”

Results

  • Figure 1: please improve the resolution and the size of the figure in order to improve the readability in particular for the panel C and F that actually are not readable at all. Moreover regarding the panel C that report the Summary table of identified or putatively 99 annotated compounds in both ionization modes  please consider to make them as a separate table in order to make it more clear
  • Figure 2: Please in both panel A and B make more clear the sign of statistical differences
  • Table 1: please specify the acronym in the footnote
  • Figure 4: Please in panel A make more clear the sign of statistical differences
  • Figure 5: please improve the resolution and the size of the figure in order to improve the readability

Discussion

The Discussion section is comprehensive, structured, and well-linked to the experimental data, providing a clear mechanistic interpretation of the observed anti-MASLD effects of EB-WE. The authors effectively connect their findings to known literature, supporting their hypotheses with appropriate citations and biochemical pathways.

However, some improvements are needed:

  • The text is repetitive in some parts (e.g., multiple mentions of AMPK roles or antioxidant effects), which could be condensed.
  • Some sentences are overly long and complex, making the discussion harder to follow (e.g., lines 403–409, 425–436).
  • Minor grammatical issues appear, e.g., missing articles ("how EB-WE ameliorate MASLD" → "how EB-WE ameliorates MASLD").
  • The section is rich in information but sometimes jumps abruptly between pathways (e.g., PGC-1α/UCP-1 to CPT-1 to SIRT without clear transition).
  • The discussion would benefit from clearer thematic grouping, e.g.: Regulation of lipid metabolism; Enhancement of insulin sensitivity; Antioxidant activity; Identification of bioactive compounds
  • Some conclusions about causality (e.g., EB-WE effects being definitively mediated by pyropheophorbide A and loliolide) are too strong given that direct bioactivity assays on isolated compounds were not yet performed.

Additionally while some limitations are briefly mentioned (e.g., need for further validation), a more explicit reflection on the limitations of compound identification, model system generalizability, and lack of dose-response analysis would strengthen scientific rigor. Therefore I suggest to add a paragraph discussing these limitations and suggesting how future studies could address them.

Materials and methods

In section 4.4. Please provide the number of ethical authorization for the present study

Author Response

(The authors gave the same response as above.)

Round 2

Reviewer 1 Report

Comments and Suggestions for Authors

The authors have revised it according to the comments of the reviewers and can accept it.

Reviewer 3 Report

Comments and Suggestions for Authors

The authors fully responded to the comments made thus making significant improvements to the manuscript which I therefore consider suitable for publication